# Pharmacokinetics of the Anti-Inflammatory Drug Meloxicam after Single 1.5 mg/kg Intramuscular Administration to Undulate Skates (*Raja undulata*)

**DOI:** 10.3390/vetsci9050216

**Published:** 2022-04-28

**Authors:** Pablo Morón-Elorza, Daniela Cañizares-Cooz, Carlos Rojo-Solis, Teresa Álvaro-Álvarez, Mónica Valls-Torres, Daniel García-Párraga, Teresa Encinas

**Affiliations:** 1Department of Pharmacology and Toxicology, Faculty of Veterinary Medicine, Complutense University of Madrid, Av. Puerta de Hierro s/n, 28040 Madrid, Spain; dcaniz01@ucm.es (D.C.-C.); tencinas@vet.ucm.es (T.E.); 2Fundación Oceanogràfic de la Comunitat Valenciana, C/Eduardo Primo Yúfera (Científic) 1B, 46013 Valencia, Spain; dgarcia@oceanografic.org; 3Veterinary Services, Oceanogràfic, Ciudad de las Artes y las Ciencias, C/Eduardo Primo Yúfera (Científic) 1B, 46013 Valencia, Spain; crojo@oceanografic.org (C.R.-S.); talvaro@oceanografic.org (T.Á.-Á.); mvalls@oceanografic.org (M.V.-T.)

**Keywords:** elasmobranch, chondrichthyan, pain, analgesia, NSAID, HLPC, drug kinetics, fish cyclooxygenase, inflammation

## Abstract

The therapy database currently used in elasmobranchs is still mostly based on empirical data, and there are few efficacy and safety studies supporting clinical practice. In this study, meloxicam pharmacokinetics (PK) were evaluated after a single 1.5 mg/kg IM administration to a group of seven clinically healthy adult undulate skates (*Raja undulata* Lacepède, 1802). Blood samples were collected before administration and at 15, 30, 60 and 90 min and 2, 4, 8, 12, 24 and 48 h after the IM injection. The meloxicam concentrations in plasma were determined using high-performance liquid chromatography, and PK parameters were calculated using a non-compartmental model approach. The mean ± SEM values of the main PK values were 1.84 ± 0.31 μg/mL for peak plasma concentration, 1.5 ± 0.24 h for time to maximum plasma concentration, 11.43 ± 2.04 h·µg/mL for area under the plasma concentration vs. time curve, 3.55 ± 0.65 h for elimination half-life, and 5.37 ± 0.94 h for mean residency time. No adverse reactions were detected. The relatively high plasma concentration and short time to maximum plasma concentration suggest that meloxicam could turn into an efficient analgesic and anti-inflammatory candidate drug to be used in skates. Further efficacy, pharmacodynamic, and multiple-dose studies with meloxicam are needed in elasmobranchs.

## 1. Introduction

Meloxicam is a cyclooxygenase (COX)-2 selective non-steroidal anti-inflammatory drug (NSAID) that is currently indicated and successfully used in the therapeutical management of a wide range of companion and exotic animal species, due to its great selectivity for COX-2, good absorption through oral (PO) and intramuscular (IM) routes, and wide safety margins [1,2]. However, pharmacokinetics (PK) of meloxicam have shown great variations among the species studied to date (mainly mammals, avian and reptile species), with significant differences in the dosages required to achieve clinically effective plasma concentrations and important variations in half-life and clearance [1,3]. In all the studied species, once meloxicam is absorbed, it is metabolized mainly in the liver via cytochrome p450 (CYP) enzymes, transformed into four inactive metabolites and eliminated in feces and urine [4,5]. Previous studies provide evidence that pharmaceutical and xenobiotic-metabolization via CYP enzymes, including NSAIDs, should be carefully studied before extrapolation across taxa, as fish CYPs have proven to respond differently compared to mammals [6]. These differences are important, as they can influence the speed at which some drugs are metabolized and eliminated, affecting clearance rates and leading to variations in the kinetic properties or toxic effects of those drugs that undergo liver metabolization [7,8]. PK studies aim to examine how the concentration of a drug evolves over time, determining the amounts of that drug and its metabolites in the tissues of different species, their fluids, and excrements [9]. In addition, PK studies also help the clinician to adjust the drug dosages and inter-dosage periods and allows the prediction of the results obtained during a pharmacological treatment, since pharmacological response is determined by the ability of the drug to access the sites of action and drug concentration over time [10]. PK behavior can show great variations between species, even if these species are very close phylogenetically [3].

There is currently very little information regarding pain and inflammation management in fish. The drug concentrations reached at their sites of action are not known for most of the drugs used, with much of the veterinary clinical practice depending on the extrapolation of data from other species and in most cases, lacking the corresponding PK, pharmacodynamic (PD), efficacy and safety studies [11,12]. There are few published PK studies performed with meloxicam in fish. In teleosts, meloxicam has been only recently studied after intravenous (IV), IM and PO routes in rainbow trout (*Oncorhynchus mykiss* Walbaum, 1792) and Nile tilapia (*Oreochromys niloticus* Linnaeus, 1758) [12,13,14]. In addition, although meloxicam is frequently prescribed in the clinical management of elasmobranchs maintained in aquariums and marine rehabilitation centers, it has been studied only in nursehounds (*Scyliorhinus stellaris* Linnaeus, 1758) and yellow stingrays (*Urobatis jamaicensis* Cuvier, 1816) [15,16,17,18]. Given the high prevalence of infectious and inflammatory diseases in elasmobranchs and the frequent prescription of meloxicam in this group of animals, this study sought to determine the kinetic properties of meloxicam when administered IM to undulate skates and to provide information that could help improve the therapeutic management of pain and inflammation in elasmobranchs and most specifically in skates [17,19,20].

## 2. Materials and Methods

### 2.1. Animals

The study population consisted of seven adult (three females, four males) undulate skates (*Raja undulata* Lacepède, 1802), with weights ranging from 3.20 kg to 4.5 kg (mean ± SD was 3.94 ± 0.41 kg) and a mean total length ± SD of 76.67 ± 5.64 cm. All animals were maintained in the exhibit tanks of the Oceanogràfic Aquarium of Valencia for a minimum period of four years. For the purpose of the study, the skates were transferred to a temporary 10,000-L cylindrical tank at the quarantine facilities of the aquarium, to allow better access and management of the animals during the study. The undulate skates were translocated to the experimental tanks 14 days prior to the onset of the PK study to allow them to acclimate to their new environment. The seven individuals were determined clinically healthy based on their medical history, a detailed physical examination, hematology, and plasma biochemistry results obtained prior to the PK study. Due to a previous study evaluating the possible effects of meloxicam on hematological and plasma chemical values during an equivalent PK trial administering meloxicam IM at 1.5 mg/kg to nursehounds, which did not reveal significant variations in blood analytical values associated with meloxicam administration, and in an effort to minimize animal handling as well as reducing the blood volume collected, this study did not perform hematological and plasma chemical follow ups in the skates [15]. However, all animals were visually monitored throughout and one month after the PK study for possible adverse reactions.

To recreate the experimental conditions used at Oceanogràfic to evaluate meloxicam PK in the other elasmobranch species, the tanks contained processed sea water collected from the Mediterranean at 17–18 °C, with a 34 g/L salt concentration and pH of 7.8–8.1. Ammonia, nitrite, and nitrate concentrations were always under 0.01, 0.05 and 50 ppm, respectively. The light cycle was a 12 h light: 12 h dark artificially controlled period, and the air temperature ranged from 15 to 24 °C. The undulate skates were fed once daily, six days per week, with thawed pieces of cephalopods and teleosts provided *ad libitum* [15].

The procedures involving animals were in compliance with the Consensus Author Guidelines on Animal Ethics and Welfare for Veterinary Journals, the EU Directive 2010/63/EU and Spanish RD 53/2013 for animal experiments and were approved by the Animal Care and Welfare Committee at Oceanogràfic of Valencia and the Generalitat Valenciana, under the project reference ID OCE-22-19.

### 2.2. Experimental Design

This study was designed as a prospective experimental trial. All the animals were closely monitored throughout the course of the study to detect possible alterations or adverse effects that could have been produced by the drug or the sampling procedure. All the individuals were uniquely identified during the study using colored plastic beads, which were placed on the edge of the disc using a surgical suture.

All the skates were weighted using a crane scale and received the same (1.5 mg/kg) intramuscular dosage at 8:00 a.m., using meloxicam (5 mg/mL Metacam^®^ solution for injection, Boehringer Ingelheim, Barcelona, Spain). The mean volume ± SD of meloxicam administered was 1.18 ± 0.12 mL. For drug administration, the animals were captured using a rubber net, and meloxicam was administered in the caudal third of the disc musculature dorsally, using a 23-gauge needle attached to a 2 mL syringe. The blood samples consisted of 0.4 mL of whole blood and were collected from each animal before drug administration and at the following times after the meloxicam injection: 15, 30, 60 and 90 min and 2, 4, 8, 12, 24 and 48 h. The total blood volume collected during the study was 4.4 mL for each individual, which represented 0.11% of the mean body weight, and was under the maximal 1% body weight total blood volume collection recommended in fish [21]. For blood collection, the animals were captured using a rubber net and placed in dorsoventral recumbency for the induction of tonic immobility; manually restrained with their body submerged in water and the tail was brought to the surface while blood was collected. Venipuncture was performed from the caudal blood vessels at the proximal ventral tail, using a 23-gauge needle attached to a 2 mL syringe [22].

As in the previous PK studies performed with elasmobranchs at Oceanogràfic, once blood was collected, it was directly placed into 1 mL lithium-heparin tubes, which were refrigerated at 4 °C and sent to the laboratory located at the veterinary clinic of the aquarium for further processing, within 30 min from collection. The blood tubes were then placed in an Ortoalresa^®^ Digicen21 CE110 centrifuge with a swing bucket rotor (Ortoalresa^®^ RT106 Na 170007/01, 132 mm rotor radius, 35-degree angle fixed) and centrifuged at room temperature (24 °C) for 5 min and 590× *g*. After centrifugation, whole plasma was collected using a micropipette, introduced into a 1.5 Eppendorf tube, and frozen at −20 °C until meloxicam quantification.

### 2.3. Meloxicam Quantification

The plasma samples were analyzed for meloxicam quantification within 30 days after collection. The meloxicam plasma concentrations were determined by high-performance liquid chromatography (HPLC) with UV detection, by using a previously described HPLC method successfully used in elasmobranchs [16]. Briefly, 250 μL of the thawed plasma sample was mixed with 100 μL of hydrochloride acid solution (5M), vortexed for 2 min, before adding 3 mL of diethyl ether and being vortexed again for 10 min. The solution was centrifuged at 3150× *g* (4500 rpm) and 4 °C for10 min using a refrigerated centrifuge (Hettich Universal 32R, Hettich Iberia, Torrejón de Ardoz 28850, Madrid, Spain), equipped with a swing bucket rotor of 140 mm rotor radius. The organic layer was collected and evaporated to dryness under a vacuum at 45 °C. The residue was resuspended in 250 μL of methanol, and 20 μL were injected in the HPLC.

Sample processing and chromatographic assays were performed at the Department of Pharmacology, Faculty of Medicine at the University of Valencia (Av. de Blasco Ibáñez, 15, 46010 València, Valencia, Spain), with a Shimadzu HPLC system (PC-FIHGUV-09, Shimadzu Europe GmbH, Albert-Hahn-Str. 6–10, D-47269 Duisburg, Germany), with a SCL-10Avp controller, SIL-10ADvp autoinjector, FDV-10ALvp quaternary pump and SP10ADvp UV detector set at 355 nm and Shimadzu LCMS Solutions software for data processing and peak integration. A C18 column (Mediterranean Sea C-18 column, Teknokroma, Barcelona, Spain), was equipped to the system. The validation of the HPLC methods was performed prior to assay, and calibration curves were generated using methanol spiked with meloxicam (Sigma-Aldrich Química SA, Tres Cantos, Madrid, Spain), displaying linear absorbance at the studied concentrations (R^2^ > 0.99). The limit of detection was 0.02 μg/mL, the limit of quantification was 0.09 μg/mL, and inter- and intra-assay variability were under 6%. The mean ± SD recovery for meloxicam in undulate skate plasma was 90.31 ± 7.40%.

### 2.4. Data Analysis

The maximal concentration in plasma (C_max_) and time to maximal concentration in plasma (T_max_) were directly determined from the plasma concentration vs. time data. The PK parameters, including elimination half-life (t_1/2__β_), area under the plasma concentration–time curve to the last sampling time (AUC_t_), area under the plasma concentration–time curve extrapolated to infinite (AUC_inf_) and mean residency time (MRT), were determined following a noncompartmental analysis, using a commercially available software (PK Solutions, version 2.0, Summit Research Services, Montrose, Colorado, USA). The elimination rate constant (K_e_) was estimated via semilog-linear regression of the terminal slope (λ_z_), based on data points automatically estimated using the regression with the largest adjusted R^2^. Elimination half-life (t_½β_) was estimated by ln2/K_e_. In this study, the AUC_t_ and area under the moment curve to the last sampling time (AUMC_t_) were determined via log-trapezoidal integration, and the AUC_inf_ was estimated based on the last observed or predicted concentration, divided by λ_z_; MRT was calculated by dividing the AUC_t_ by the AUMC_t._

## 3. Results

No adverse reactions or pathological signs were detected in any of the animals during the study, or in the following two months of its completion.

The individual meloxicam plasma concentrations after a single intramuscular administration of meloxicam at 1.5 mg/kg in *R. undulata* are represented in Figure 1. The plasma concentration vs. time curves were very similar for all the individuals administered with the drug, and the inter-individual variations were minimal. The meloxicam mean plasma concentrations (n = 7) are represented in Figure 2. Meloxicam was rapidly absorbed from the muscle in the seven individuals, with a mean T_max_ ± SEM of 1.50 ± 0.24 h and achieved high concentrations in plasma, with a mean C_max_ ± SEM of 1.84 ± 0.31 μg/mL. The mean AUC_t_ ± SEM was 11.43 ± 2.04 h·µg/mL and the AUC_inf_ ± SEM was 11.63 ± 2.08 h·µg/mL. Elimination was fast, although slightly more progressive than absorption, with a mean t_1/2__β_ ± SEM of 3.55 ± 0.65 h and mean MRT ± SEM of 5.37 ± 0.94 h. The PK parameters of meloxicam after 1.5 mg/kg IM administration are represented in Table 1.

## 4. Discussion

To the authors’ knowledge, despite the frequently diagnosed inflammatory diseases that affect elasmobranchs and the frequent prescription of meloxicam in this group of animals, meloxicam PKs had never been studied in skates. So far, studies have only been conducted in two elasmobranch species, the nursehound shark and the yellow stingray [17,18,19]. The drug was easily administered to the undulate skates, and no adverse reactions were observed after the administration of 1.5 mg/kg IM. Previous PK studies with meloxicam at 0.5 mg/kg and 1.5 mg/kg IM and PO in nursehounds and 1mg/kg IM and 2 mg/kg PO in yellow stingrays, did not produce any detectable adverse effects, and dosages as high as 5 mg/kg meloxicam IM have been safely administered to goldfish, without evidencing toxic effects [15,16,26].

The intramuscular administration of meloxicam at 1.5 mg/kg in undulate skates resulted in a fast and effective absorption, with a mean C_max_ of 1.84 μg/mL, which was similar to that reported for Nile tilapia (1.95 μg/mL) and yellow stingrays (1.29 μg/mL) after administering meloxicam at 1 mg/kg IM, although higher than the C_max_ registered in previous studies with nursehounds (0.81 μg/mL) after giving meloxicam at 1.5 mg/kg IM [14,15]. The C_max_ was also high when compared to previous studies performed with different mammalian, avian and reptile species [14]. The relatively higher C_max_ would suggest that clinically effective concentrations could be achieved in undulate skates using the described treatment protocol. The fast absorption with a short T_max_ (1.5 h) could be useful for short and moderately painful procedures that could lead to inflammation, such as biopsy collection or minor surgeries.

Meloxicam elimination was also fast in the undulate skates, with a relatively short t_1/2__β_ (3.55 h) when compared to other mammalian species, such as rabbits (6.1 h) or horses (8.5 h) [14,27]. The elimination half-life was, however, within range of that reported in teleost species, such as tilapia (1.59 h) and trout (4.55 h), reptile species, such as the loggerhead sea turtle (*Caretta caretta* Linnaeus, 1758) (3.26 h), and avian species, such as the zebra finch (*Taeniopygia guttata* Vieillot, 1817) (3.24 h) [12,14,28,29]. Previous studies performed with meloxicam in avian species showed that interspecific and interindividual variations in Phase I system enzymes, such as cytochromes P450 (CYPs), can result in longer half-lives of metabolized NSAIDs [7]. CYPs are important proteins involved in pharmaceutical metabolism; they are frequently studied and well defined in mammals, although their function in most non-mammalian vertebrates is much less understood [6]. The differences in Phase I enzymes may be linked to the highly variable PK parameters among fish. A previous study evaluated the metabolizing system of deep-sea fish using several CYP isoforms and showed species-specific differences in baseline activities and sensitivity to different chemicals, including the NSAID diclofenac [8]. These variations could be leading to the greater clearance of meloxicam in some fish species and enhance the importance of developing further PK and PD studies in fish, as the required dosages and therapeutic protocols can result in great variations among species.

Interestingly, the t_1/2__β_ was shorter in undulate skates (t_1/2__β_ = 3.55 h) when compared to yellow stingrays (t_1/2__β_ = 5.75 h) after 1 mg/kg IM administration and nursehounds (t_1/2__β_ = 15.97 h) after meloxicam 1.5 mg/kg IM administration, showing important variations in elimination for meloxicam among elasmobranch species [15,18]. The three studied elasmobranch species belong to different orders (undulate skates belong to order *Rajiformes*, yellow stingrays to order *Myliobatiformes* and nursehounds to order *Carcharhiniformes*), and there are many factors that could have produced the previously mentioned differences in the PK parameters, as anatomical, physiological, and environmental peculiarities can greatly influence the behavior of the drugs administered to fish [30,31]. Previous studies showed that the enzymatic metabolism of the small-spotted catshark (*Scyliorhinus canicula* Linnaeus, 1758) was lower than that of teleost fish [32]. These differences in metabolism and elimination could lead to the maintenance of high meloxicam concentrations in plasma for longer periods in some elasmobranch species compared to teleost, and therefore to much more efficient administration protocols for this drug in sharks. For this reason, it is especially important to determine the pharmacokinetic profiles in each group of animals, and to carefully assess the possibility of extrapolating therapeutic regimens from other species.

Most elasmobranchs and teleost fish are poikilothermic, with their body temperature (and therefore the enzyme activity and metabolism, drug clearance and duration of effect) dependent on the environmental water temperature [11,33]. Because of this, environmental conditions, such as water temperature, salinity and pH, could greatly influence drug kinetics and should be recorded during a PK study involving fish [30]. In our study, the undulate skates were kept at the same environmental conditions as the catsharks used in the preliminary meloxicam PK studies (18.0 °C, 34 g/L, ph 8, pO2 > 95%), while Nile tilapia were kept at 23.3–24.4 °C during the meloxicam PK trial [14]. The yellow stingrays were kept at higher water temperatures (24.0–28.0 °C) and showed a longer t_1/2__β_ than undulate skates, although shorter than nursehounds. Further PK studies using elasmobranchs maintained at different temperatures and salinities could provide important information on how these environmental factors influence meloxicam kinetics in fish.

Great variations have been reported in the standard metabolic rates (SMR) among the elasmobranch species [34]. Meloxicam elimination in undulate skates was faster when compared to nursehounds, which could initially suggest that undulate skates have a higher SMR that those of nursehounds [15]. However, previous studies performed with nursehounds and the little skate (*Raja erinacea* Mitchill, 1825) (which belongs to the same genus as the *Raja undulata* and has a similar distribution and benthic behavior), reported similar SMR for both species [34,35]. These results would imply that the nursehounds and undulate skates in our studies, being both benthic species, healthy adult animals, and maintained at the same environmental conditions, had similar SMRs, despite the great variations in the following elimination PK parameters: a mean t_1/2__β_ of 3.55 h and MRT of 5.37 h in undulate skates and mean t_1/2__β_ of 15.97 h and MRT of 23.40 h in nursehounds, after meloxicam 1.5 mg/kg IM administration in both cases [15]. These results would suggest that there are other important factors that affect meloxicam kinetics in elasmobranchs, such as protein binding capacity or differences in metabolization and excretion, which should be evaluated.

Within elasmobranchs, the size and composition of the liver depend on the species, which, together with the amount of liver cell exposure, can greatly influence the pharmacodynamics and kinetics of the drugs [20]. In addition, fish kidneys are very different from those of mammals; they have a higher filtration rate and different filtration, secretion, and reabsorption selectivity [36]. These differences can influence the elimination rates of the drugs such as meloxicam, which is partly eliminated in urine [37]. Previous studies also revealed the following kidney differences between freshwater and marine fish: marine species have generally smaller and fewer glomeruli, together with lower filtration rates to those of freshwater fish species [38,39]. The differences in kidney function between te freshwater and marine fish species could be influencing meloxicam elimination, and further PK studies with meloxicam in marine teleost species would help to understand the observed differences in t_1/2__β_ between the studied fish species. Although kidney-related side effects of meloxicam have been described in mammalian and avian species, they are considered milder and less frequent when compared to other NSAIDs, due to meloxicam’s weak inhibitory activity on COX-1 [2,3]. Regarding teleost, a previous acute toxicity study provided evidence that a single intramuscular injection of meloxicam at a dosage of 5 mg/kg does not cause acute toxicity, nor significant histological kidney alterations, determining meloxicam safe for administration to goldfish at the studied dosage [26]. Despite the fact that no clinical signs were detected in the undulate skates and nursehounds after meloxicam administration, and due to the absence of published NSAID toxicity studies in elasmobranchs, further single and long-term meloxicam administration toxicity studies are needed in this group of animals [15,16]. The biochemical processes in elasmobranchs also present very notable differences compared to the rest of vertebrates, many of them related to their high urea retention index; the peculiar metabolic organization and the unique functionality of biological membranes can decisively interfere with the kinetic behavior of multiple substances and lead to important differences between elasmobranchs and other vertebrate species [40]. Finally, the renal portal system could have also affected drug kinetic parameters in all the studied fish species. In an effort to replicate the most frequently indicated injection sites in fish therapeutics, meloxicam was administered in the epaxial musculature, latero-caudal to the dorsal fin in nursehounds, tilapia and trout, and in the disc musculature of the undulate skates [12,14,15,16]. These regions belong to the caudal half of the animals and could have been affected by the renal portal system, as the system allows blood from this region to drain directly to the kidneys; this could have increased pre-systemic renal excretion and modified drug distribution [36]. Additional PK studies are granted to evaluate the differences in meloxicam kinetics after administration in the cranial-third of the body in fish, to discard a possible renal portal system interaction with drug distribution and elimination.

The determination of the meloxicam clinical effective concentration in undulate skates was not the objective of the present study, and in the absence of studies determining the effective meloxicam plasma concentrations required for COX-2 inhibition in fish, further studies investigating the pharmacodynamics of meloxicam in rays and sharks are required. The mean meloxicam plasma concentration at 8 h after administration was 0.51± 0.04 μg/mL in undulate skates, which is over the 0.39 μg/mL meloxicam plasma concentration described in dogs for analgesic/anti-inflammatory effects [41]. In addition, the mean plasma concentration at 12 h post-administration was 0.25 ± 0.03 μg/mL, which is still over the mean inhibitory concentration of COX-2 described in other animal species, such as horses (0.13 to 0.19 μg/mL), potentially leading to clinically relevant concentrations for longer periods [23]. However, the current therapeutical management of fish tries to minimize handling as much as possible, in an effort to reduce stress and the risk of trauma related to animal capture for drug administration [14]. In this sense, inter-dosage periods requiring handlings as frequent as every 8 h, although achievable in small and resilient species, such as the undulate skate, could be too frequent for some easily stressed elasmobranch species, and could be counterproductive [11]. Prolonged release meloxicam injectable formulations have provided a longer t_1/2__β_ in mammalian vertebrates, compared to the conventional injection solution used in this study [42]. Although prolonged release meloxicam formulations are still not evaluated in elasmobranchs, given the important differences observed in PK parameters, the good absorption and fast elimination after IM administration, further studies with prolonged release formulations together with viability studies could result in longer inter-dosage periods, which could benefit the therapeutical management of some elasmobranchs and teleost fish species.

## 5. Conclusions

The results provided in this study suggest that meloxicam administered intramuscularly at a dosage of 1.5 mg/kg in undulate skates was rapidly absorbed, metabolized, and excreted. The studied administration could provide clinically relevant plasma concentrations for 8–12 h, although further clinical efficacy and PD studies are needed to better determine the potential of meloxicam as an analgesic and anti-inflammatory drug to be used in undulate skates.

This study also identifies the possible factors that could be related to the important differences observed in meloxicam PK parameters between the different teleost and elasmobranch species, revealing the limited information available in the field of fish pharmacology and therapeutics, while opening new prospects for further research lines, which could significantly improve pain management and inflammation therapeutics in these species.

## Figures and Tables

**Figure 1 vetsci-09-00216-f001:**
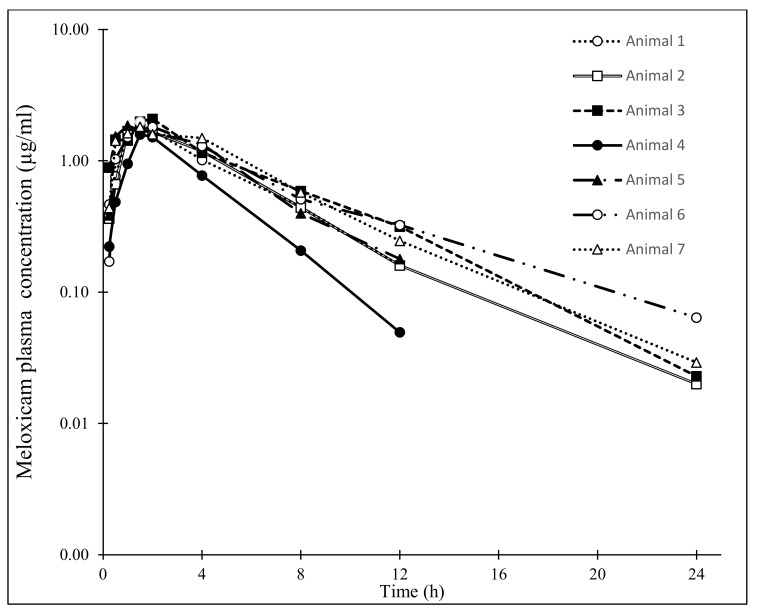
Individual plasma concentrations of meloxicam in undulate skates (*Raja undulata*) (n = 7) after a single IM administration (1.5 mg/kg). Please note the very similar kinetic curves among the different individuals. Meloxicam plasma concentration at 24 h in animals 1, 4 and 5 was under the limit of detection (0.02 µg/mL); concentration at 48 h was under the limit of detection in all individuals.

**Figure 2 vetsci-09-00216-f002:**
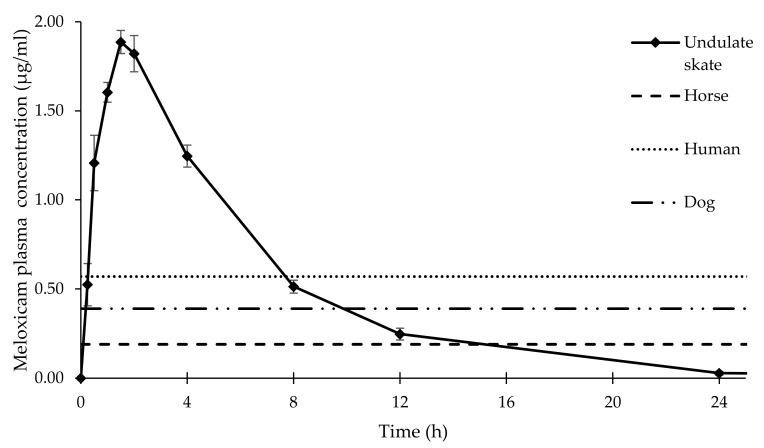
Mean plasma concentrations ± SEM of meloxicam 1.5mg/kg IM in undulate skates (*Raja undulata*) (n = 7). Reference meloxicam concentrations required for anti-inflammatory effect in different animal species are also represented [23,24,25]. Please note that mean meloxicam concentrations are maintained over inhibitory concentrations (IC) determined for other animal species for 8–12 h after IM administration. Meloxicam plasma concentration at 48 h was under the limit of detection in all individuals.

**Table 1 vetsci-09-00216-t001:** Pharmacokinetic parameters of meloxicam in seven undulate skates (*Raja undulata*) (n = 7) after a single intramuscular administration (1.5 mg/kg).

Parameter (Unit)	MEAN	SEM
T_max_ (h)	1.50	0.24
C_max_ (μg/mL)	1.84	0.31
t_1/2__β_ (h)	3.55	0.65
AUC_t_ (h·µg/mL)	11.43	2.04
AUC_inf_ (h·µg/mL)	11.63	2.08
AUC_t/inf_ (h·µg/mL)	0.98	0.16
MRT (h)	5.37	0.94

T_max_ = time to maximum concentration; C_max_ = maximum plasma concentration; t_½β_ = terminal half-life; AUC_t_ = area under the curve until last sampling; AUC_inf_ = area under the curve extrapolated to infinity; MRT = mean residency time.

## Data Availability

Data obtained in this study are available upon request to the corresponding author.

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
