# Peer review of "Pharmacokinetics of the Anti-Inflammatory Drug Meloxicam after Single 1.5 mg/kg Intramuscular Administration to Undulate Skates (Raja undulata)"

_vetsci, 2022, doi:10.3390/vetsci9050216_

Round 1

Reviewer 1 Report

This study evaluates the pharmacokinetic parameters of intramuscular meloxicam in undulate skates. Given the very limited information available about elasmobranch species, and the frequent need for analgesia, this study provides valuable information to clinicians caring for this species.

The study design is sound, and the authors do not overstate the conclusions or recommendations. I have a few comments and suggestions below:

Line 159. Consider further describing the formulas/equations that the software used to analyze the model and calculate the parameters.

Line 170. You mentioned that no adverse reactions were noted during or after the study, and previously mentioned that they were deemed clinically healthy based on exam and bloodwork parameters. Were you able to follow up with hematology/biochemistry results either during or after the study, and if so, were there any clinically significant changes? (If you were not able to follow up with bloodwork, perhaps because of sample volume, consider mentioning this and reason.)

Figure 2. Consider re-labeling the legend 'meloxicam' with 'skate' or 'R. undulata.' While I understand that the diamond points are showing the concentrations and the other dashed lines show reference concentrations in other species, it is a bit confusing to read 'meloxicam', 'horse', human, etc. 

[General comment: consider changing 'levels' to 'concentrations' for consistency. Both words are used interchangeably.]

Line 298. I'm glad you mentioned the renal portal system as a potential source for altering PK parameters. And while you did not see any clinical changes, given that a significant side effect of meloxicam in other species is renal impairment, consider mentioning how nephrotoxicity could occur with meloxicam administration (or if it has or hasn't been reported with meloxicam in fish).

Author Response

Thank you very much for your detailed corrections, which allowed us to improve our work. Explanations to the corrections made are stated in blue after each of your comments:

Line 159. Consider further describing the formulas/equations that the software used to analyze the model and calculate the parameters.

We appreciate your suggestion. A sentence has been included in the Data analysis section (Line 171) further describing how parameters were calculated.

Line 170. You mentioned that no adverse reactions were noted during or after the study, and previously mentioned that they were deemed clinically healthy based on exam and bloodwork parameters. Were you able to follow up with hematology/biochemistry results either during or after the study, and if so, were there any clinically significant changes? (If you were not able to follow up with bloodwork, perhaps because of sample volume, consider mentioning this and reason.)

Thank you very much for the important commentary. A previous study already evaluated variations in hematology and plasma chemistry values during an equivalent pharmacokinetic trial with meloxicam administered intramuscularly at 1.5 mg/kg to nursehounds maintained under the same environmental conditions. Due to the absence of significant variations associated with meloxicam in the mentioned study, as well as the absence of adverse reactions and clinical signs detected both in sharks and skates, a follow up with bloodwork was not performed in this study in an effort to reduce animal handlings and collected blood volume to the minimum. A detailed explanation has been provided in materials and methods (Lines 85-92).

Reference:

Morón-Elorza, P.; Rojo-Solís, C.; Álvaro-Álvarez, T.; Valls-Torres, M.; García-Párraga, D.; Encinas, T. Pharmacokinetics of Meloxicam after Single 1.5 Mg/Kg Intramuscular Administration to Nursehound Sharks (Scyliorhinus stellaris) and Its Effects on Hematology and Plasma Biochemistry. [Accepted for publication 5th February 2022]; J. Zoo Wild. Med. 2022, 53, doi:10.1638/2021-0144.

Figure 2. Consider re-labeling the legend 'meloxicam' with 'skate' or 'R. undulata.' While I understand that the diamond points are showing the concentrations and the other dashed lines show reference concentrations in other species, it is a bit confusing to read 'meloxicam', 'horse', human, etc. 

Thank you for the excellent suggestion. The legend has been re-labeled according to your recommendation.

[General comment: consider changing 'levels' to 'concentrations' for consistency. Both words are used interchangeably.]

We appreciate it. The world “levels” has been revised to “concentrations” throughout the manuscript.

Line 298. I'm glad you mentioned the renal portal system as a potential source for altering PK parameters. And while you did not see any clinical changes, given that a significant side effect of meloxicam in other species is renal impairment, consider mentioning how nephrotoxicity could occur with meloxicam administration (or if it has or hasn't been reported with meloxicam in fish).

Thank you very much for the commentary.  Due to its weak inhibitory action on COX-1, meloxicam's adverse effects are considered to be milder and less frequent than other non-selective NSAIDs, and meloxicam appears to be safe in mammals even in the presence of chronic kidney disease (Gowan RA et al. 2012). Meloxicam has also been demonstrated to have very few adverse effects in birds even with severe overdosing (Summa NM et al. 2017)

To our knowledge, there is only one published study evaluating the toxicity of meloxicam in fish, which administered a single injection of 5 mg/kg intramuscularly to healthy goldfish (Carassius auratus auratus) (Larouche CB et al. 2018). Though further studies are needed to evaluate the toxicity of long-term meloxicam administration in fish, the mentioned study provided evidence that a single IM injection of meloxicam at a dosage up to 5 mg/kg does not cause acute toxicity nor significant alterations in the kidneys during histological examination in goldfish. This would be a very interesting field of research to develop in chondrichthyans, as there are still no toxicity studies with NSAIDs in this large group of fish and no published long-term toxicity studies with NSAIDs in fish. A statement reflecting this point has been included in the discussion section of the manuscript (Lines 326-332).  

Reviewer 2 Report

Dear Authors, 

the manuscript "vetsci-1693774" focuses on an interesting topic, highlighting pharmacokinetics in the elasmobranch Raja undulata (Lacepède, 1802).

The paper is well written and, despite present few data, it is the first evaluation of pharmacokinetics in this elasmobranch species. 

I have just few minor suggestions to improve the quality of the manuscript.

  • In the keywords please avoid the use of terms already used in the text
  • each first time a species is mentioned please be sure to indicate species authorities e.g. Raja undulata (Lacepède, 1802)
  • Maybe "Meloxicam quantification" section should be numbered

    in figure 2 labeling please change "meloxicam" with "Raja undulata"

Author Response

Thank you very much for your detailed corrections, which allowed us to improve our work. Explanations to the corrections made are stated in blue after each of your comments:

In the keywords please avoid the use of terms already used in the text

Thank you for the commentary. Keywords have been revised to avoid using terms already included in the title.

Each first time a species is mentioned please be sure to indicate species authorities e.g. Raja undulata (Lacepède, 1802)

Thank you. We have included species authorities throughout the manuscript.

Maybe "Meloxicam quantification" section should be numbered

We appreciate your suggestion. The section has been numbered.

In figure 2 labeling please change "meloxicam" with "Raja undulata"

Thank you, “meloxicam” has been revised to “Undulate skate” in figure 2 according to your commentary and to reviewer #1 suggestion.

Reviewer 3 Report

This is an excellent manuscript on the pharmacokinetics of meloxicam after a single IM dose in undulate skates (Raja undulata). The research design and data analysis are appropriate, and the Results and Discussion is acceptable based on the data. The only problem with the manuscript is some of the English word usage, most of which are not commonly used in the authors’ context.

Specific corrections/edits the authors should consider in a revision include the following:

Lines 40, 172, 175, 195, 313, 314, 316 – replace “plasmatic” with just “plasma”.

Lines 43, 227, 233, 271 – replace “former” with “previous”.

Line 77 – replace “at” with “in”.

Line 92 – “ab libitum” should be in italics.

Line 94 – replace “are” with “were”.

Line 115 – delete the word “greatly” as this is a relative term without specifics.

Line 120 – replace “at” with “from”.

Line 128 – “590 g” should be “590 x g”

Line 139 - “315 g” should be “315 x g”

Line 139 - replace “during” with “for”.

Line 184 – “Undulate” should be lower cased.

Line 216 – replace “high” with “higher”.

Line 221 – replace “small” with “minor”.

Line 238 – replace “can suffer” with “can result in”

Line 344 – replace “can” with “could”

Line 364 – both “Aquarist” and “Veterinary” should be lower cased.

Author Response

Thank you very much for your detailed corrections, which allowed us to improve our work. The manuscript has been revised following your corrections/edits.